# A PIANO (Proper, Insufficient, Aberrant, and NO Reprogramming) Response to the Yamanaka Factors in the Initial Stages of Human iPSC Reprogramming

**DOI:** 10.3390/ijms21093229

**Published:** 2020-05-02

**Authors:** Kejin Hu

**Affiliations:** Department of Biochemistry and Molecular Genetics, School of Medicine, University of Alabama at Birmingham, Birmingham, AL 35294, USA; kejinhu@uab.edu

**Keywords:** human induced pluripotent stem cells, human iPSC, reprogramming legitimacy, aberrant reprogramming, transcriptional profiling, reprogramome, transcriptional response, Yamanaka factors, metabolic reprogramming

## Abstract

Yamanaka reprogramming is revolutionary but inefficient, slow, and stochastic. The underlying molecular events for these mixed outcomes of induction of pluripotent stem cells (iPSC) reprogramming is still unclear. Previous studies about transcriptional responses to reprogramming overlooked human reprogramming and are compromised by the fact that only a rare population proceeds towards pluripotency, and a significant amount of the collected transcriptional data may not represent the positive reprogramming. We recently developed a concept of reprogramome, which allows one to study the early transcriptional responses to the Yamanaka factors in the perspective of reprogramming legitimacy of a gene response to reprogramming. Using RNA-seq, this study scored 579 genes successfully reprogrammed within 48 h, indicating the potency of the reprogramming factors. This report also tallied 438 genes reprogrammed significantly but insufficiently up to 72 h, indicating a positive drive with some inadequacy of the Yamanaka factors. In addition, 953 member genes within the reprogramome were transcriptionally irresponsive to reprogramming, showing the inability of the reprogramming factors to directly act on these genes. Furthermore, there were 305 genes undergoing six types of aberrant reprogramming: over, wrong, and unwanted upreprogramming or downreprogramming, revealing significant negative impacts of the Yamanaka factors. The mixed findings about the initial transcriptional responses to the reprogramming factors shed new insights into the robustness as well as limitations of the Yamanaka factors.

## 1. Introduction

Human pluripotent stem cells (PSCs) can be generated from somatic cells, generally fibroblasts, by ectopic expression of the four Yamanaka factors, OCT4, SOX2, KLF4, and c-MYC (collectively known as OSKM) [1]. OSKM induction of PSCs (iPSCs) is revolutionary and powerful, but very inefficient, stochastic, and slow [2] compared to the reprogramming using the natural reprogramming device, oocytes [3]. A very small population of the starting fibroblasts (<1%) becomes iPSCs over a long period of time (>2 weeks) in a stochastic manner [4,5,6,7]. The underlying molecular events to account for this mixed outcome are still poorly understood. The Yamanaka reprogramming is so revolutionary and has amazed scientists so much that previous studies of iPSC reprogramming via transcriptional profiling have focused on what happens during the reprogramming process, assuming implicitly that most, if not all, of the transcriptional responses to the Yamanaka factors were positive. Those studies did not consider the legitimacy of a gene response to the reprogramming factors, and their analyses are compromised by noise from 99% of the cells that do not proceed towards the direction of pluripotency, especially the data from the early or initial stages of reprogramming [8]. To mitigate the impact of the noise signals, some laboratories compared transcriptional profiles for a few purified populations positive for a very limited number of surface markers during the reprogramming process with that of the starting cells [9], but those populations are still highly heterogeneous. The insights gained from the previous studies are also limited by a lack of innovative concepts in their analyses of the profiling data. In addition, previous such studies predominantly focused on the mouse reprogramming [8,9,10,11] and may not represent the molecular events in the human pluripotency reprogramming since human PSCs are very different in morphology, culture conditions, cell surface marker profile, and differentiation potentials, and human iPSC reprogramming is much slower, and even much less efficient [6,7,12].

Recently, we proposed a novel concept of reprogramome, which is the full complement of genes that have to be reprogrammed for a successful conversion of cell fates from one type to another [13]. Using reprogramming of human fibroblasts into iPSCs as a model, we have previously defined the full complement of genes that need to be reprogrammed to the levels of expression found in the embryonic stem cells (ESC), i.e., human fibroblast-to-iPSC reprogramome. This concept allows one to decide the legitimacy of a gene response to the reprogramming factors. The concepts of reprogramome and reprogramming legitimacy remove flaws in previous studies. The interpretation of the previous data is compromised without taking into consideration of reprogramming legitimacy of a gene response to reprogramming. For example, the proper upreprogramming for the pluripotent surface gene *PODXL* for both mouse and human [14] was mistakenly regarded as an aberrant and unwanted upreprogramming because this gene is best known as a regulator for the terminally differentiated kidney podocytes and is less known as a pluripotency signature gene to many scientists [11]. Following our previous defining of the human fibroblast-to-iPSC reprogramome, this study further evaluates the legitimacy of the transcriptional responses of genes to the Yamanaka reprogramming. The current study has scored various types of transcriptional responses to the conventional OSKM reprogramming. These include successful, insufficient, and no reprogramming, as well as six types of aberrant reprogramming. The current observations can explain very well the robustness, as well as the significant limitations, of the Yamanaka reprogramming. The cataloged genes of the different types of reprogramming outcomes provide a starting point for future investigations and improvements of the iPSC technology.

## 2. Results

### 2.1. Genes in Nucleic Acid Metabolism and Ribosome Biogenesis Are Successfully Reprogrammed to the Pluripotent State within 48 h

iPSC reprogramming is revolutionary, but it is inefficient, stochastic, and slow. To understand why Yamanaka factors can reprogram fibroblasts to pluripotency but do so inefficiently, stochastically, and slowly, we need to find out how many, if any, and what kind of genes are successfully reprogrammed, and which genes are not reprogramed in the initial stages. We also need to know if there is any aberrant reprogramming that can account for the intrinsically inefficient, stochastic, and slow natures of the Yamanaka reprogramming.

To answer these questions, this study took advantage of the novel concept of reprogramome we proposed recently [13]. This new concept allows one to define two sets of genes that need to be reprogrammed. The first set is the pluripotency-enriched genes, i.e., upreprogramome, which have to be upreprogrammed to the expression levels found in the human PSCs. The second set of genes is the human fibroblast-enriched genes, i.e., downreprogramome, which have to be downregulated to the levels found in PSCs. This study first generated a more rigorous upreprogramome and downreprogramome by using more stringent criteria (see Materials and Methods), and the two resulting sub-reprogramomes serve as the references for the analyses in this report (see Appendix A for the upreprogramome and downreprogramome, respectively).

Albeit of low efficiency, Yamanaka factors can convert a rare population of the starting cells into iPSCs. To account for this success, I hypothesized that a significant amount of genes in the reprogramome may have been properly reprogrammed already at the initial stages. To test this, this project used the unbiased comprehensive RNA-seq technology to sequence RNA from the reprogramming cells at the very early stages. To define the genes that have a consistent and reliable early transcriptional response to the Yamanaka factors, we sequenced RNA from two distinct time points but still at the same very early reprogramming stage, i.e., 48 h and 72 h post factor transduction, and considered only those genes that have the same response at both time points. All of the four factors were successfully overexpressed at both early time points (48 h and 72 h) (Appendix A and Appendix A). Combining the two time points, OSKM induced differential expression of 2480 genes when compared with the naïve human fibroblasts, and 2791 genes when compared with the fibroblasts transduced with lentiviral green fluorescent protein GFP (Appendix A). For the same time points of reprogramming, this report identified 1801 more differentially expressed genes than a previous similar study (2480 vs. 679 genes) even though the current work used more stringent criteria (2×, *q* < 0.01 vs. 1.5×, *q* < 0.05) (Appendix A) [15]. Two major reasons account for this discrepancy: (1) the profiling method here is the highly sensitive, unbiased, and comprehensive genome-wide RNA-seq, and Mah et al. used microarray [15], which is limited by probe sets and low sensitivity; (2) they used gamma retroviruses to transduce human fibroblasts, while the current study used lentiviral vectors to deliver the OSKM transgenes. Retroviral transduction efficiency of human cells is very low [4], and in fact, the efficiency of their retroviral transduction was only 27.6% while we have routinely reached >90% of transduction of human fibroblasts with our lentiviral vectors [6,7,14,16].

Overexpression of GFP mediated by the lentiviral vector impacted a small group of genes compared with that of OSKM overexpression (see below), upregulating 141 genes, and downregulating 10 genes only (Appendix A). To eliminate the GFP/virus effects, albeit small, in all the following results, this study just considers genes that responded to OSKM in the same way in both the naïve human fibroblasts and those transduced with GFP viruses so that the changes in gene expressions truly represent responses to the OSKM reprogramming factors.

Upon overexpression of OSKM, 946 genes were commonly upregulated in both naïve fibroblasts and fibroblasts transduced with the GFP viruses considering both time points (Supplementary Figure 1D). Impressively, 366 of those 946 genes have been properly upreprogrammed to the levels found in human embryonic stem cells (ESCs) (Figure 1A,B and Appendix A), representing 38.7% of the upregulated genes by OSKM. To understand the nature of these properly reprogrammed genes, gene ontology (GO) analyses were conducted. Surprisingly, 238 of the 362 mapped genes out of the 366 genes successfully reprogrammed by OSKM (65.8%) have the common GO term of “metabolic process” (GO #, 0009987) (Figure 2). Further examination of the GO data revealed that the majority of the reprogrammed metabolic genes are involved in nucleic acid metabolism (115 out of the 238 genes, i.e., 48.3% of cellular metabolism genes mapped), 83 of which are involved in RNA metabolism (72.2% of nucleic acid metabolism) (Figure 3, Appendix A). Of note, among the 83 RNA metabolic genes reprogrammed within 48 h, the majority are genes involved in ribosome biogenesis (54 out of 83, i.e., 65.1% of the reprogrammed RNA metabolic genes mapped). Finally, 44 out of the 54 reprogrammed ribosome biogenesis genes have roles in rRNA processing (GO #, 0006364) (81.5%) (Appendix A and Appendix A). In agreement with the observation here, a previous proteomic profiling of reprogramming also showed that proteins in “RNA processing” were strongly induced at the early stage of mouse iPSC reprogramming but without considering the reprogramming legitimacy of these RNA-processing genes [17]. In summary, OSKM quickly reprogram RNA metabolic genes, especially rRNA processing genes into the pluripotent state.

### 2.2. A Set of Promiscuous Fibroblast Genes and Genes for Focal Adhesion and Vacuole of Cellular Components Are Quickly Downreprogrammed to the Pluripotent State

To further account for the robustness of the Yamanaka factors, the next question is whether any gene is successfully downreprogrammed at the early stage to the levels found in PSCs with the downreprogramome as a reference. Similar to upreprogramming, it is found that a large set of genes (213 genes) have been successfully downreprogrammed to the levels found in PSCs (Figure 4 and Appendix A). Of note, the expression of a small group of genes was efficiently shut off by OSKM within 48 h, and those genes are not expressed in PSCs, for example, *GDF5*, *GPR39*, *IDNK*, *XDH*, *SLC9A9*, and *P2RX7* (Figure 4C and Appendix A). It is noticed that the expression of this group of silenced genes by OSKM is at low levels in the naïve fibroblasts, but the expressions are substantial; for example, the lowest of them, *P2RX7*, has an average read count of 112.8. There was no shutoff of gene expression for any of the differentially expressed genes that have high and higher expression levels in the starting fibroblasts. Some genes with extremely high expression in fibroblasts were efficiently reprogrammed to the pluripotent levels such as FTL (from read counts 194,299 to 41,055) and FTH1 (from read counts 114,956 to 30,642) (Figure 4C and Appendix A).

GO analyses were next conducted for the successfully downreprogrammed genes (213 genes). In dramatic contrast to the properly upreprogrammed genes, there were no statistically significant results for the overrepresentation tests with the databases of “GO biological process complete”, “GO molecular function complete”, “PANTEHR pathways”, and “Reactome pathways”. These results indicate that, unlike the successfully upreprogrammed genes, the genes successfully downreprogrammed are more promiscuous. Nevertheless, a test with the “GO cellular component complete” returned two small groups of genes with statistical significances, “anchoring junction” and “vacuole” (Figure 5A). The first group is mainly of focal adhesion with 19 genes identified (Figure 5A–D and Appendix A). Among the properly downreprogrammed are 21 genes of vacuole components (Appendix A and Appendix A).

### 2.3. Two Sets of Genes Were Significantly but Insufficiently Reprogrammed at the Early Stages

The above analyses indicate that OSKM successfully reprogram 579 genes to the levels in PSCs (366 up and 213 down) within 48 h, indicating the robustness of the Yamanaka reprogramming factors. However, Yamanaka reprograming is very inefficient (<1%), slow, and stochastic. To account for the limitation of Yamanaka reprogramming, the next question asked is what happens to the remaining genes in the reprogramome. In the upreprogramome, it is found that 152 genes were significantly upregulated but insufficiently reprogrammed. The expression level for each of the 152 ESC-enriched genes is still at least 2-fold lower than that in ESCs, although each of those genes has been upregulated at least 2-fold (Figure 6A, Appendix A, Appendix A). Because of these two criteria, any gene in this category is at least 4-fold enriched in expression in ESCs compared to that in the somatic fibroblasts. We previously reported that the pluripotent surface marker gene *PODXL* is quickly activated upon reprogramming [14]. The new data here are in agreement with our previous finding, but *PODXL* was still significantly insufficiently reprogramed because it was upregulated by 18.4-fold to reach an average normalized read count of 4197, but its expression is still 11.3 times lower than that in PSCs.

GO analysis of these 152 genes with the “biological process complete” database returned only one statistically significant GO term of “DNA replication” with 12 genes (*CHTF18*, *DSCC1*, *DTL*, *CCNE1*, *ORC1*, *ATAD5*, *DBF4*, *MCM3*, *CHAF1A*, *GINS1*, *CDC6*, and *TICRR*). GO analyses resulted in no statistically significant results for “molecular function complete”, “Reactome pathways”, and “PANTHER pathways”. However, GO analyses with “cellular component complete” showed that genes involved in cytoskeleton and cell junction were overrepresented significantly (Appendix A). These two cellular components are functionally related, and there were nine common members between these two groups (*AIF1L*, *CGN*, *CTTNBP2*, *EPB41L4B*, *PNN*, *PODXL*, *PPP1R9A*, *SLC16A1*, and *SPTBN2*).

In the downreprogramome, 286 genes were significantly downregulated but insufficiently downreprogrammed by OSKM (Figure 6B, Appendix A, Appendix A, see Materials and Methods for defining this set of genes). For example, we previously reported that *SNAI2*, *PSG5*, *NR2F2*, and *IL1R1* were robustly expressed in human fibroblasts but not expressed in human PSCs [18]. This is confirmed by the current data here. These fibroblast genes were downregulated 5- to 17-fold by OSKM within 48 h, but these genes are still not shut off as required because their expression levels were still at least 6.6 to 40.6 times above the threshold (50 normalized read counts).

Unlike the genes insufficiently upreprogrammed and those successfully downreprogrammed, GO analyses of the 286 significantly downregulated but insufficiently downreprogrammed genes result in a long list of GO terms overrepresented significantly (Appendix A). Of note, a set of genes involved in development were among this list. Similarly, a set of genes with roles in morphogenesis and differentiation were also insufficiently downreprogrammed at this early stage (Appendix A). These data indicate that significant reprogramming on those cellular features (e.g, downregulation of developmental genes and of regulators of the somatic cell morphology) has occurred but further reprogramming is needed.

### 2.4. Two Large Sets of Highly Enriched and Robustly Expressed Genes Did Not Respond to Reprogramming Factors at the Initial Stages

Yamanaka reprogramming is inefficient, slow, and stochastic; one hypothesis to account for these limitations is that many genes are not responsive to OSKM reprogramming at the early stages of reprogramming. In testing this hypothesis, this study mainly focused on genes with higher expression (the mean normalized read count is >500) and great differences between the starting cells and the endpoint PSCs (at least 5-fold differences in expression levels) (see Materials and Methods for defining this group of genes) in an assumption that these genes may be more critical than those with low expression and fewer differences in expression levels.

In the upreprogramome, 504 out of the 1033 highly PSC-enriched and well-expressed genes were not responsive to the reprogramming factors, representing 48.8% of the robustly expressed and PSC-enriched genes (see Materials and Methods, Figure 7A, and Appendix A, Appendix A). Of note, many well-established pluripotency genes were resistant to reprogramming. For example, *NANOG, LIN28A, LIN28B, TERT, TERF1, LEFTY1, ZFP42, DNMT3B, DPPA4, PRDM14, SALL4*, and *ZSCAN10* were all transcriptionally refractory to OSKM induction within the first 72 h.

Like the significantly downregulated but insufficiently downreprogrammed genes, the irresponsive PSC-enriched genes are not random and not unrelated. GO analyses with the “biological process complete” database showed that 66 GO terms were significantly overrepresented (FDR < 0.05), and 37 of them were overrepresented at least 2-fold. Among these PSC-enriched genes irresponsive to OSKM reprogramming are those involved in embryo development and stem cell population maintenance (Appendix A). Like genes significantly downregulated but insufficiently downreprogrammed, a lot of the PSC-enriched irresponsive genes have roles in cell morphogenesis; for example, 45 PSC-enriched irresponsive genes were associated with the GO term of “cell morphogenesis”, and 48 such genes were associated with the GO term of “cellular component morphogenesis”. The PSC-enriched irresponsive genes were also overrepresented in the groups with roles in cell adhesion and cell junction, with 49 such genes associated with the GO term of “cell adhesion” (Appendix A). Impressively, the reactome-pathways analysis showed that genes for “transcriptional regulation of pluripotent stem cells” were 18.7-fold overrepresented in this irresponsive group with 10 of the 22 members for this reactome pathway being in the PSC-enriched irresponsive group, including *NANOG, EPHA1, TDGF1, DPPA4, ZSCAN10, FOXD3, LIN28A, PRDM14, ZIC3*, and *SALL4* (data not visualized, but can be found in the interactive Appendix A). This observation is in agreement with the previous work on mouse reprogramming in that they reported that many pluripotency key regulators were activated very late, i.e., in the stabilization phase, for example, mouse *Lin28* and *Dppa4* [8]. Proteomic profiling also showed that mouse Lin28, Sall4, and Dppa4 were expressed only at a very late stage of reprogramming [17]. The data here are probably an underestimation because two members of this reactome pathway (*OCT4* and *SOX2*) likely do not respond to OSKM induction initially and the current data cannot reveal the responses of these two transcriptional regulators of pluripotent stem cells because they were among the four reprogramming factors overexpressed in the RNA-seq samples in this report. Indeed, previous research showed that murine *Oct4* [19] and *Sox2* were activated only late during reprogramming [9,20]. In the partially reprogrammed mouse iPSC lines, the endogenous *Oct4* is not activated and the Oct4 protein is the product of the integrated *Oct4* transgene [10]. Human endogenous *SOX2* was also not detectable before 72 h of fibroblast reprogramming [15]. In summary, these data indicate that pluripotency cell morphology, cell adhesion, and cell–cell interaction, as well as pluripotency transcriptional network are among the strongest barriers to reprogramming at the initial stages.

Next, the author tested whether any gene in the downreprogramome is also resistant to reprogramming as seen in the upreprogramome. Indeed, 449 of the 1129 highly fibroblast-enriched and robustly expressed genes were not responsive to OSKM reprogramming within the first 72 h, representing 39.8% of this group (Figure 7B, Appendix A, Appendix A). For examples, the current RNA-seq data confirmed our previous microarray data that *EMP1, LOX,* and *NFIX* were robustly expressed in human fibroblasts but not expressed or expressed at very low levels in human PSCs [18] and showed that they were not responding transcriptionally to OSKM reprogramming at the initial stage.

The author next wondered what the functions for these irresponsive fibroblast-enriched genes are. The initial GO analysis with the “biological process complete” database returned as many as 251 overrepresented GO terms Focus was then shifted to the more evolutionally conserved “biological process-slim” database, and it was found that 43 evolutionally conserved GO terms were overrepresented by this 449-gene set (Appendix A). Again, genes involved in cellular morphogenesis and cell shape were overrepresented by the irresponsive fibroblast genes and seven morphogenesis GO terms were associated with these genes, including 19 genes playing roles in “cell morphogenesis” (Appendix A). Of note, many GO terms for cellular behaviors were overrepresented in the irresponsive fibroblast gene set; for examples, cell spreading (6 genes, overrepresented 12.2-fold), regulation of cell shape (7 genes, overrepresented 7-fold), cell migration (20 genes), cell motility (20 genes), cell adhesion (24 genes), cell communication (30 genes), cell development (19 genes), cell differentiation (36 genes), cell localization (20 genes), and cell population proliferation (12 genes) (Appendix A). These results indicate that genes that maintain the somatic fibroblast cellular morphology and cellular behaviors constitute significant barriers to reprogramming at the initial stages of iPSC reprogramming.

### 2.5. Six Types of Aberrant Reprogramming

To identify additional factors contributing to the inefficient, slow, and stochastic reprogramming by the Yamanaka factors, it was hypothesized that some genes might be wrongly reprogrammed at the early stage of reprogramming. That is, some genes are wrongly upregulated at least 2-fold when they should be downreprogrammed at least 2-fold, or some genes are wrongly downregulated at least 2-fold while they should be upreprogrammed at least 2-fold.

It was found that 22 genes were wrongly upregulated at least 2-fold, where they should be downreprogrammed at least 2-fold (Figure 8A, Appendix A, Appendix A). On the other hand, 26 genes were wrongly downregulated at least 2-fold where they should be upreprogrammed at least 2-fold (Figure 8B, Appendix A, Appendix A). In these two cases, the expression gap to be closed by reprogramming became even greater, and was increased to the extent of at least 4-fold.

There are two other possibilities: some genes are upregulated at least 2-fold where there are no differences in expression levels between the starting somatic cells and the endpoint PSCs, i.e., unwanted upreprogramming; on the other hand, there may be unwanted downreprogramming of genes which are downregulated at least 2-fold, but there are no differences in expression levels between the starting cells and the endpoint pluripotent cells. Indeed, 134 genes were identified in the unwanted upreprogramming category (Figure 8C, Appendix A, Appendix A) and 99 genes in the unwanted downreprogramming group (Figure 8D, Appendix A, Appendix A).

Finally, 18 genes were over-upreprogrammed in that these PSC-enriched genes were upregulated to the levels at least 2-fold higher than that in PSCs and at least 4-fold higher than that in the starting fibroblasts (Figure 8E, Appendix A, Appendix A). Lastly, six genes were over-downreprogrammed to a similar extent (*CXCL12*, *NR1H3*, *VEPH1*, *TMEM35A*, *MRVI1*, *PKD1L2*) (Appendix A).

GO analyses were conducted with each set of the six types of aberrantly reprogrammed genes individually but resulted in no statistically significant GO terms with all the six sets of genes except for two cases. GO analysis of the 99 unwanted downreprogrammed genes with the PANTHER pathways indicated that the “metabotropic glutamate receptor group II pathway” was 17.6-fold overrepresented (four genes were in this group vs. 0.23 expected genes. They are *CACNA1A*, *VAMP1*, *STX1B*, and *GNAO1*.). In the second case, GO analysis of the 134 unwanted upreprogrammed genes with the “cellular component complete” annotation data set indicated that the GO term of “extracellular space” was overrepresented by 1.95-fold (42 observed genes vs. 21.48 expected. Data not shown). Then, all of the 305 genes aberrantly reprogrammed were pooled as a single list to run PANTHER GO tests. No statistically significant results were found for any of the GO annotation data sets, indicating that the aberrant reprogramming is more promiscuous. Function classification was conducted for the 305 genes in this group. These genes were represented in various GO classes within different classifications including protein class, cellular component, molecular function, and biological process (Appendix A). Of note, 57 membrane genes were aberrantly reprogrammed, 23 of which were organelle membrane proteins. Twelve of these membrane proteins have transporter activities. Another interesting family of proteins in this aberrant reprogramming group is those playing roles in metabolisms.

## 3. Discussion

iPSC reprogramming has been studied intensely over the past 14 years; however, it remains a black box to us. This technology is still inefficient, slow, stochastic, and error-prone. There is a possibility that the efficiency and fidelity can be enhanced significantly considering that the oocyte reprogramming is fast, efficient, and accurate [3]. To better understand and study the pluripotency reprogramming, we recently proposed a novel concept of reprogramome, which is the full complement of genes that have to be reprogrammed to the levels found in the pluripotent stem cells. The concept of reprogramome allows one to determine the legitimacy of a gene response to the reprogramming factors. Previous mouse research identified many patterns of transcriptional changes by comparing expression profiles of reprogramming populations purified with only two markers, one somatic and one intermediately marker SSEA1 [9], but those transcriptional responses do not necessarily represent the required transcriptional changes for successful reprogramming considering the very low efficiency and the stochastic nature of the iPSC reprogramming methods, as well as the high heterogeneity both in gene expression and reprogrammability within the SSEA1^+^ population. The noise signals are even greater for the previous mouse temporal expression profiling at different reprogramming stages of the unsorted secondary reprogramming MEF harboring the pre-integrated inducible OSKM reprogramming factors because the reprogramming efficiency is still very low with that system and the cells are very heterogeneous [8]. With consideration of the legitimacy of the transcriptional responses to reprogramming, the current study has cataloged many different types of transcriptional responses of genes to the conventional Yamanaka factors. Compared to the previous analyses, the patterns of responses described here better explain the robustness and limitations of the Yamanaka reprogramming.

Furthermore, the previous research predominantly focused on mouse reprogramming. The current report fills in a gap as the detailed study of early transcriptional responses to OSKM in human pluripotency reprogramming. One work reported the transcriptional responses to OSKM in human pluripotency reprogramming [15]. However, that investigation was limited by its use of the insensitive and biased microarray profiling. Critically, the authors used retroviruses to deliver the four reprogramming factors into human fibroblasts, and only 27.6% of the cells were transduced. The current research is more comprehensive, complete, and accurate thanks to the use of RNA-seq and our optimized transduction protocols. As a result, the current study identified at least 3.6× more differentially regulated genes by OSKM. Furthermore, a significant portion of the regulated genes in Mah’s data may reflect the impact of viruses on the reprogramming cells since their responsive genes were predominantly overrepresented by GO terms of “response to virus”, “immune response”, and “oxidative stress”. Of note, this study excluded the viral impact by using the viral GFP as controls. Indeed, the GO terms of “response to virus”, “immune response”, and “oxidative stress” did not appear in the upregulated gene list here, indicating that the transcriptional responses identified here represent the true responses to OSKM, not to the viruses.

The current research focuses on the very early stage of reprogramming because the early stages encounter the initial and critical barriers and also provide a solid foundation for some rare cells to pass the reprogramming threshold and proceed to pluripotency. Indeed, it was found that a large set of 579 genes has been successfully reprogrammed already within the first 48 h, including 366 pluripotency-enriched genes successfully upreprogrammed to the pluripotency levels and 213 somatic genes successfully downreprogrammed to the pluripotency state. The successful reprogramming of the 579 genes at the very early stage constitutes the early molecular underpinnings for the potency of the OSKM reprogramming. This study further cataloged 438 genes that were significantly but insufficiently reprogrammed up to the point of 72 h. These include 152 pluripotency-enriched genes and 286 fibroblast-enriched genes. The mixed results in this category explain both the potency and the limitation of the OSKM reprogramming.

Impressively, 953 genes were resistant to OSKM reprogramming up to the point of 72 h. These include the 504 pluripotency-enriched genes. In agreement with the previous studies, many pluripotency signature genes are among this category, for example, *NANOG*, *LIN28*, *LEFTY1*, *PRDM14*, *SALL4*, *ZSCAN10*, *DPPA4,* and *DNMT3B*. Critically, GO analysis indicated that more than 10 genes in the reactome pathway for transcriptional regulation of the pluripotent stem cells were not responsive to OSKM reprogramming. The current data are in agreement with the previous observations that the major pluripotency-associated genes are reprogrammed at the last stage [9]. Inability of OSKM to directly reprogram these critical pluripotency signature genes and the transcriptional regulators of pluripotency may be a major barrier in the current iPSC protocol.

Among the irresponsive genes are 449 fibroblast-enriched genes. Previous study showed that erasure of somatic transcription is an early event. However, the current data indicate that downreprogramming of somatic genes still constitutes a great barrier to iPSC reprogramming. This is not surprising since it was reported that there is some somatic memory in pluripotency reprogramming [21,22]. GO analyses indicate that these refractory somatic genes are not random, and a lot of them have roles in cellular morphogenesis and various cellular behavior such as spreading, migration, adhesion, communication, motility, differentiation, localization, and development of cells. These results indicate that the somatic cell morphology, shape, and behaviors are refractory to OSKM reprogramming.

Unexpectedly, there were six types of aberrant reprogramming: wrong upreprogramming, wrong downrerpogramming, unwanted upreprogramming, unwanted downreprogramming, over-upreprogramming, and over-downreprogramming. Together, this broad group includes 305 genes. The aberrant reprogramming of OSKM seems random or promiscuous since no statistically significant GO term for these 305 genes was found. The promiscuous nature of the aberrant reprogramming is reasonable because OSKM are pluripotent regulators, and the promiscuous outcome may be that these factors are over-expressed during reprogramming, especially at the early time of reprogramming. This aberrant reprogramming with 305 genes may constitute a major limitation of the OSKM reprogramming.

We previously reported that genes involved in cellular morphogenesis have to be extensively and intensely reprogrammed for a successful conversion of human fibroblasts to iPSCs [13]. The current study reveals that these genes fall into three different categories. First, a set of genes for fibroblast morphology has been significantly downregulated but insufficiently reprogrammed (Appendix A), indicating a partial success of morphogenetic reprogramming. Second, another set of genes for fibroblast morphology is resistant to reprogramming (Appendix A), indicating significant reprogramming block in this cellular feature. Third, a set of genes for PSC morphology is resistant to reprogramming, and there is no morphogenesis GO term among the 140 overrepresented GO terms for the 366 successfully upreprogrammed pluripotency genes. These results are in agreement with our laboratory experience in that fibroblasts change their morphology to some degree at the early stage but pluripotency morphology appears at a later stage. These results also indicate that both loss of fibroblast morphology and establishment of pluripotent morphology are barriers of early reprogramming, although the latter is a greater barrier.

Figure 9 summarizes findings about the positive and negative roles and inability of the OSKM reprogramming factors during the initial stages of iPSC reprogramming. The successfully reprogrammed 579 genes constitute the strongest molecular underpinnings for the robustness of OSKM reprogramming. The partially reprogrammed 438 genes contribute positively to the initial success of iPSC reprogramming with some limitations. The 953 refractory genes to initial reprogramming are the greatest barrier at the early reprogramming. This is particularly true for the refractory pluripotency signature genes and the transcriptional regulators of pluripotency. The 305 aberrantly reprogrammed genes are the greatest derailing and defiance of reprogramming.

The current work has cataloged genes positively responding to OSKM reprogramming, and also those refractory to reprogramming as well as those aberrantly impacted by OSKM. This provides a starting point for further investigations of iPSC reprogramming and improvements of the technology.

## 4. Materials and Methods

### 4.1. Cells and Cell Culture

Human primary fibroblasts were purchased from ATCC (BJ, CRL-2522, Manassas, Virginia, state abbreviation if USA or Canada, country, USA), and were cultured in the fibroblast medium: Dulbecco’s Modified Eagle Medium with high glucose, supplemented with 10% heat-inactivated fetal bovine serum, 0.1 mM 2-mercaptoethanol, 100 U/mL penicillin and 100 μg/mL streptomycin, 0.1 mM MEM Non-Essential Amino Acids, and 4 ng/mL human FGF2.

Human embryonic stem cell lines H1 and H9 (WiCell, Madison, Wisconsin, USA) were cultured in the feeder-free chemically defined E8 media. The E8 medium is composed of the base media DMEM/F-12 supplemented with 1 g/L sodium chloride, 1.74 g/L NaHCO_3_, 64 mg/L L-ascorbic acid 2-phosphate sesquimagnesium, 13.6 µg/L sodium selenium, 10 μg/mL transferrin, 20 μg/mL insulin, 2 μg/L TGFβ1, and 8 ng/mL FGF2, pH 7.4, with an osmolarity of 340 mOsm adjusted with NaCl.

The lentiviral reprogramming factors were packaged in the Lenti-X 293T cells (ClonTech, Moutain View, CA, USA). The Lenti-X 293T cells were maintained in the 293T growth medium: the base media DMEM supplemented with 10% FBS, 1× MEM Non-Essential Amino Acid (MEM NEAA), 100 U/mL penicillin, and 100 μg/mL streptomycin.

### 4.2. Viral Transduction of Human Fibroblasts

The lentiviral reprogramming vectors were packaged using calcium/phosphate precipitation method, and the resulting virus particles were concentrated with PEI precipitation and subsequent centrifugation. The titers of the concentrated viruses were determined using flow cytometry taking advantage of the GFP co-expressed with each reprogramming factor as described before [6,7,14,16].

Human fibroblast BJ cells (2 × 10^5^ cells per treatment) were transduced with the four Yamanaka factors (OCT4, SOX2, KLF4, and c-MYC) at an multiplicity of infection (MOI) of 5 for each factor in the presence of polybrene at 4 μg/mL. The residual viral particles were removed 12 h post-transduction by a medium change. The transduced cells were grown in fibroblast medium till RNA harvesting.

### 4.3. RNA Sequencing

Total RNA was extracted using the TRIzol Reagent (Ambion, Austin, TX, USA) at the indicated time points. The quality of the total RNA was initially assessed by Nanodrop analyses and then further analyzed by the Agilent 2100 Bioanalyzer. The RNA was selected twice for the polyA^+^ population before conversion to cDNA using the oligo dT-magnetic beads.

We use the in-house Illumina HiSeq2500 to sequence the mRNA using the sequencing reagents and flow cells providing up to 300 Gb of sequence information per flow cell. The stranded mRNA library generation kits were used per manufacturer’s instructions (Agilent, Santa Clara, CA, USA). To establish the cDNA library, we randomly fragmentized the polyA mRNA and then used the random primers to generate cDNA with the inclusion of Actinomycin D in the first strand reaction. The ends of the cDNA were repaired, A-tailed, and ligated with adaptors for indexing during the sequencing runs. The cDNA libraries were quantitated by qPCR before cluster generation. Clusters were generated to yield approximately 725 to 825 K clusters per mm^2^. Cluster density and quality were determined during the run after the first base addition parameters were assessed. We ran paired-end 2 × 50 bp sequencing runs to align the cDNA sequences to the human reference genome.

### 4.4. Bioinformatics

All samples contained a minimum of 28.1 million reads with an average number of 40.5 million reads across all biological replicates. The FASTQ files were uploaded to the UAB High-Performance Computer cluster for bioinformatics analysis with the following custom pipeline built in the Snakemake workflow system (v5.2.2) [23]: initial quality and control of the reads were quantified with FastQC, and trimming of the bases with quality scores of less than 20 was executed with Trim_Galore! (v0.4.5). All samples passed initial FASTQ QC for a typical RNA-seq run. Following trimming, the transcripts were quasi-mapped and quantified with Salmon [24] (v0.12.0) to the hg38 human transcriptome from Gencode release 29. The average quasi-mapping rate was 87.2%, and run reports were summarized with MultiQC [25] (v1.6). The quasi-mapping results were imported into a local RStudio session (R version 3.5.3), and the package “tximport” [26] (v1.10.0) was utilized for gene-level summarization. Differential expression analysis was performed with DESeq2 package [27](v1.22.1).

### 4.5. Visualization of a Large Set of Data

All heat maps, ladder plots, and box plots were prepared with the log2-transformed normalized read counts in the R platform. All heat maps were prepared using the *pheatmap* package [28]. The ladder plots were prepared using the package *plotrix*. The box plots were prepared using the package of *ggplot2*. A heat map included all the individual read counts, while a box plot or a ladder plot was prepared using the averaged read counts of all repeats for a specific treatment or cell type.

### 4.6. Gene Ontology Analyses

The web-based PANTHER tools (PANTHER 15.0, www.pantherdb.org) were used for analyses of the input gene lists obtained using various criteria to sort the RNA-seq data. All GO analyses were conducted on the reference lists of *Homo sapiens* of PANTHER 15.0. The default Fisher’s Exact is used for the statistical overrepresentation tests, and the significance is determined by the calculated false discovery rate (FDR < 0.05).

### 4.7. Selection Criteria

#### 4.7.1. Reprogramome

The definitions for reprogramome and subreprogramome have been reported before [13]. In this manuscript, the genes with low and inconsistent expression were removed from the original reprogramome. For this purpose, a gene is considered as expressed in a specific cell type only when the normalized DESeq2 counts are greater than 50 for all individual treatments/repeats (four human naive fibroblasts and three human embryonic stem cell lines). This criterion is stricter than our previous practice, in which only the average normalized DESeq2 read counts were greater than 50 and the individual read counts were greater than 10 only. The rationales to use the cutoffs of 50 and 10 were described previously [13]. Because of this more stringent criterion, the modified reprogramome is slightly smaller (3808 genes in the preprogrammed, and 3366 genes in the downreprogramome)m and the genes with very low expression are more consistent and reliable than those previously defined. As a result, the factual smallest average normalized read counts for the pluripotency gene set (upreprogramome) becomes 54.3 (Appendix A), while the factual smallest average normalized read counts for the somatic gene set (downreprogramome) is 56.6 (Appendix A).

#### 4.7.2. Criteria for Differentially Expressed Genes

For a gene to be designated as the differentially expressed gene, it has to meet three criteria: (1) the difference in expression should be at least 2-fold rather than 1.5-fold used by similar research [15]; (2) the q-value (adjusted *p*-value) should be less than 0.01 rather than 0.05 used by similar research [15]. This manuscript does not use the *p*-values because it is more relaxed. Because of this, the actual greatest *p*-values are smaller than 0.01; for example, in the new upreprogramome, the greatest *p*-value is 0.00385; (3) for changes of gene expressions caused by OSKM, both comparisons with naïve fibroblasts and fibroblasts treated with GFP viruses, should meet the first two criteria.

#### 4.7.3. Defining the Significantly Regulated but Insufficiently Reprogrammed Genes by OSKM

To define the set of genes that are significantly downregulated but insufficiently downreprogrammed, a member gene should be downregulated at least 2-fold by OSKM, yet its expression level is still at least 2-fold higher than that in ESCs, but the expression of this gene is still above the expression threshold (50 read counts) at both time points of reprogramming (48 and 72 h), i.e., still actively expressed. The expression of this gene should be at least 2-fold higher in naïve somatic fibroblasts than in ESCs. Because of these criteria, the factual difference in expression of a given gene is at least 4-fold higher in fibroblasts than in ESCs.

To define the set of genes that are significantly upregulated but insufficiently upreprogrammed, a member gene should be upregulated by OSKM at least 2-fold, but its expression is still at least 2-fold lower than that in PSCs. The expression of a member gene should be at least 2-fold higher in PSCs than in the naïve starting cells of reprogramming, the human fibroblasts. Because of the above criteria, any member gene of this group is expressed actually at least 4-fold higher in PSCs than in fibroblasts.

#### 4.7.4. Defining the Highly Enriched Gene Sets

For highly enriched genes in fibroblasts or in ESCs, the differences in expression should be at least 5-fold, and the average normalized DESeq2 read counts should be greater than 500 in the enriched cell type. The highly enriched genes are considered not affected by OSKM when the expression levels do not meet the above criteria for the differentially expressed genes.

#### 4.7.5. Defining the Genes Irresponsive to OSKM Induction

For the genes that do not respond to OSKM, this manuscript focuses on those with higher expression and large differences in expression levels between PSC and the somatic cells assuming that those genes might be more critical biologically or in reprogramming. These sets of genes are designated as “highly enriched gene sets” as described above. To define the set of genes that do not respond to OSKM expression, OSKM should not cause a change in expression of a member gene by greater than 2-fold (up or down) except for those genes whose expression levels are below the threshold 50 both with and without OSKM expression, in which case the number of a fold change can be greater than 2. In addition, this study excludes the genes (both PSC-enriched or somatic cell-enriched) with low expression levels by including only genes whose normalized read counts are>500, which is at least 10× greater than the expression threshold 50. This study also ignores any gene whose expression difference between somatic cells and PSCs is less than 5-fold. Therefore, any member in these two groups is a highly enriched gene in the corresponding cell type, i.e., highly PSC-enriched or highly fibroblast-enriched.

## 5. Conclusions

Using the concept of reprogramming legitimacy by the logic of reprogramome, this study has identified a mixed PIANO (Properly, Insufficiently, Aberrantly, and NOT reprogrammed) response to the conventional reprogramming factors at the early stages of reprogramming. This study found that a large set of human genes (579) have been successfully reprogrammed within 48 h, and an additional set of genes (438) have been significantly reprogrammed albeit insufficiently within 72 h. These results indicate the robustness of the Yamanaka factors with some limitations at the initial stages. Interestingly, OSKM quickly reprogram genes of RNA metabolisms to the pluripotent state. However, this study has also revealed that a set of 953 genes in the reprogramome do not respond to the OSKM reprogramming including the key pluripotency signature genes and regulators of pluripotency. This group of genes is highly differentially expressed between the somatic fibroblasts and PSCs (at least 5-fold of differences in expression levels), and significant reprogramming is required for each of them in order to achieve successful reprogramming. This observation also reveals the inability of the Yamanaka factors to act directly on the key pluripotency genes, although OSKM are critical pluripotency regulators as a group. In addition, this current study identified six types of aberrant reprogramming at the early stages by OCT4, SOX2, KLF4 and c-MYC, revealing a significant negative impact of OSKM on pluripotency reprogramming. In conclusion, the observed initial PIANO responses to OSKM induction illustrate very well both the robustness and limitations of the Yamanaka factors.

All raw RNA-seq data associated with this study have been deposited at the Gene Expression Omnibus (GEO) database repository with the access code of GSE148158.

## Figures and Tables

**Figure 1 ijms-21-03229-f001:**
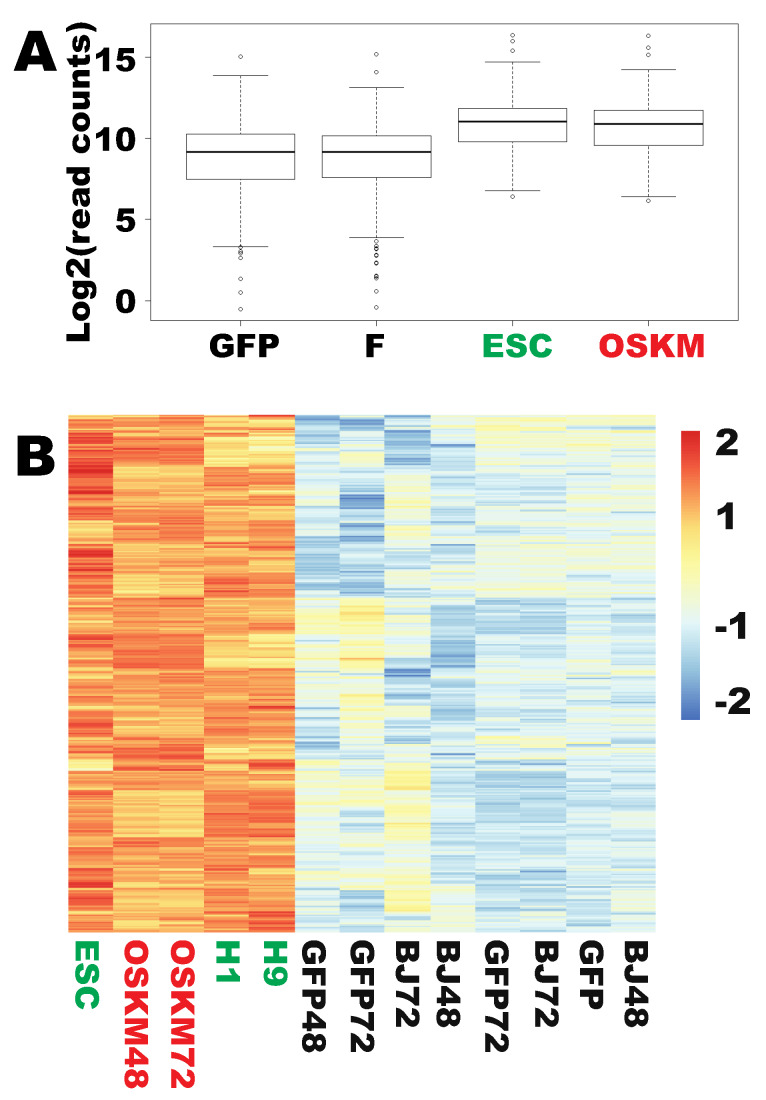
Three hundred sixty-six pluripotency-enriched genes were successfully upreprogrammed to the levels found in human ESCs. (**A**) A box plot for overall successful upreprogramming of the 366 genes at the transcriptional levels. Dots above or beneath the wiskers are outliers of each group. (**B**) A heat map showing successful individual upreprogramming of the 366 PSC genes. ESC: human embryonic stem cells (*n* = 3); H1: ESC cell line H1; H9: ESC cell line H9; OSKM: OCT4, SOX2, KLF4, and c-MYC; F: fibroblasts; BJ: human fibroblasts (*n* = 4); GFP: human fibroblasts transduced with lentiviral GFP (*n* = 4). Numbers in each label are the period of time for which transgenes had been introduced into the fibroblasts. Log2-transformed read counts were used to prepare box plots and heat maps. Heat maps were scaled by rows or genes.

**Figure 2 ijms-21-03229-f002:**
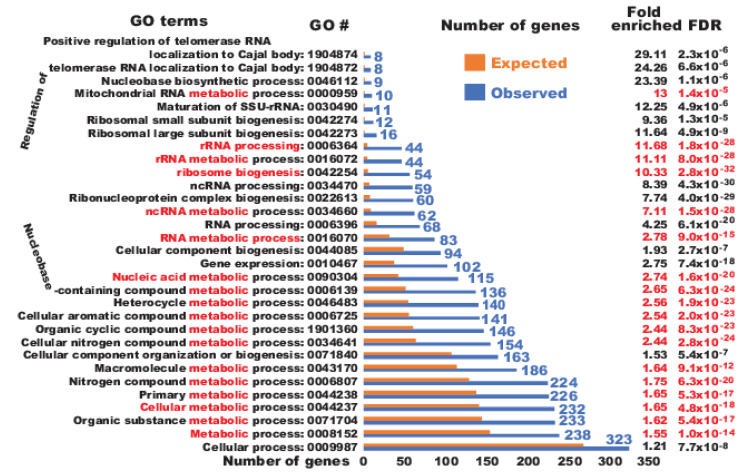
Genes involved in nucleic acid metabolisms were predominant among the early successfully reprogrammed genes. Shown is a summary of the analyses of the 366 properly reprogrammed genes with the GO annotation database of “GO biological process”. GO terms with the “metabolic” keyword and their statistical results are highlighted in red except for the numbers of genes observed for each GO term, which are all in blue. Expected numbers of genes for each GO term are not given due to space limit.

**Figure 3 ijms-21-03229-f003:**
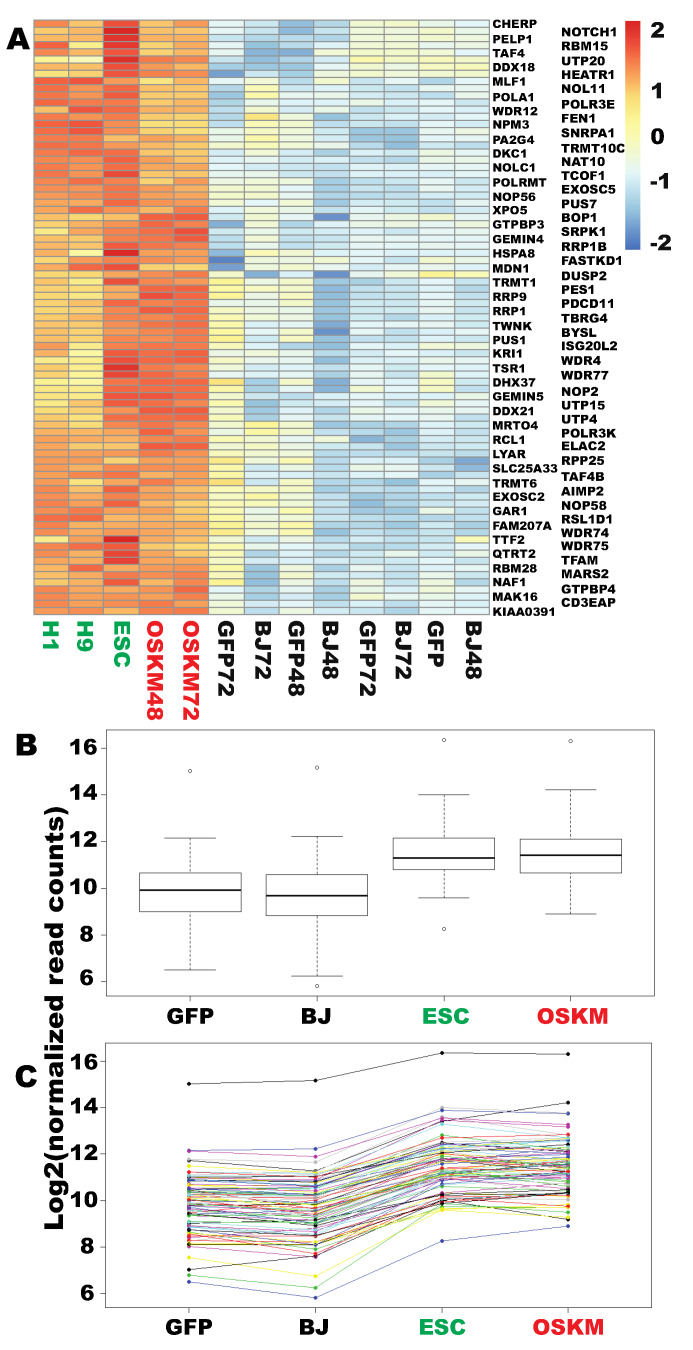
Eighty-three genes involved in RNA metabolisms were successfully reprogrammed within 48 h. (**A**) Heat map for the 83 reprogrammed RNA metabolism genes. (**B**) Box plot for the data in (**A**) to show overall successful reprogramming of those genes. Dots above or beneath the wiskers are outliers of each group. (**C**) ladder plot for the genes and their data in (**A**) and (**B**). Each line across the samples represent a specific gene. Figure sample labels are the same as in Figure 1.

**Figure 4 ijms-21-03229-f004:**
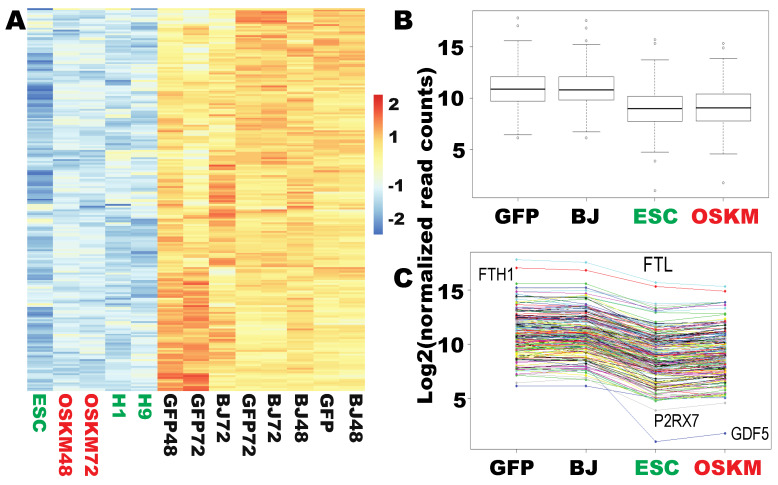
Two hundred thirteen somatic genes were successfully downreprogrammed to the pluripotency levels within the first 48 h. (**A**) Heat map of the RNA-seq data prepared as the log2-transformed normalized read counts of each sample. (**B**) Box plot of the same set of data in (**A**). Dots above or beneath the wiskers are outliers of each group. (**C**) ladder plot for data set in A and B. Each colored line across the samples represent a specific gene. Figure sample labels are the same as in Figure 1.

**Figure 5 ijms-21-03229-f005:**
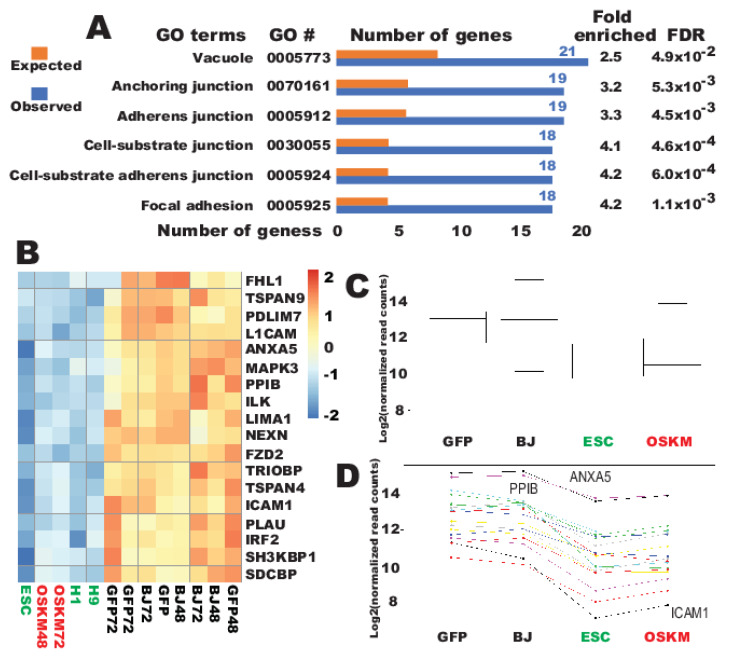
Genes for cellular components of vacuole and focal adhesion were successfully downreprogrammed by OSKM within 48 h. (**A**) Summary of GO analyses with the annotated GO dataset of “cellular component complete”. (**B**) Heat map for the 18 focal adhesion genes successfully downreprogrammed. (**C**) Box plot for data in (**B**). (**D**) Ladder plot for data in (**B**). Each colored line across the samples represent a specific gene.

**Figure 6 ijms-21-03229-f006:**
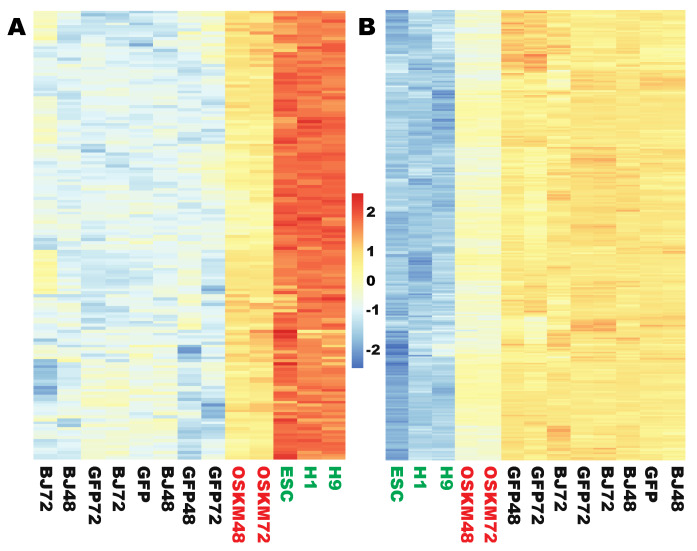
Two sets of genes were significantly but insufficiently reprogrammed by OSKM at the early stages. **(A**) A heat map showing that 152 ESC-enriched genes were significantly upregulated but insufficiently upreprogrammed by OSKM. **(B**) A heat map showing that 286 fibroblast-enriched genes were significantly downregulated but insufficiently downreprogrammed by OSKM. Figure sample labels are the same as in Figure 1.

**Figure 7 ijms-21-03229-f007:**
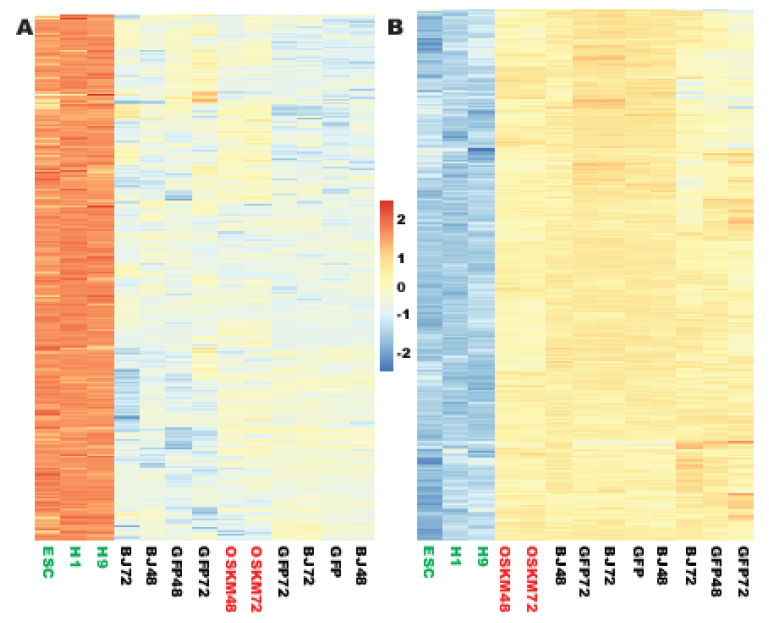
Two sets of cell-type-enriched, well-expressed genes were not responding to OSKM reprogramming at the early stage. (**A**) A heat map showing that 504 ESC-enriched genes were transcriptionally irresponsive to OSKM reprogramming at least before 72 h. (**B**) A heat map showing that 449 fibroblast-enriched genes were transcriptionally irresponsive to OSKM reprogramming at least before 72 h. Figure sample labels are the same as in Figure 1.

**Figure 8 ijms-21-03229-f008:**
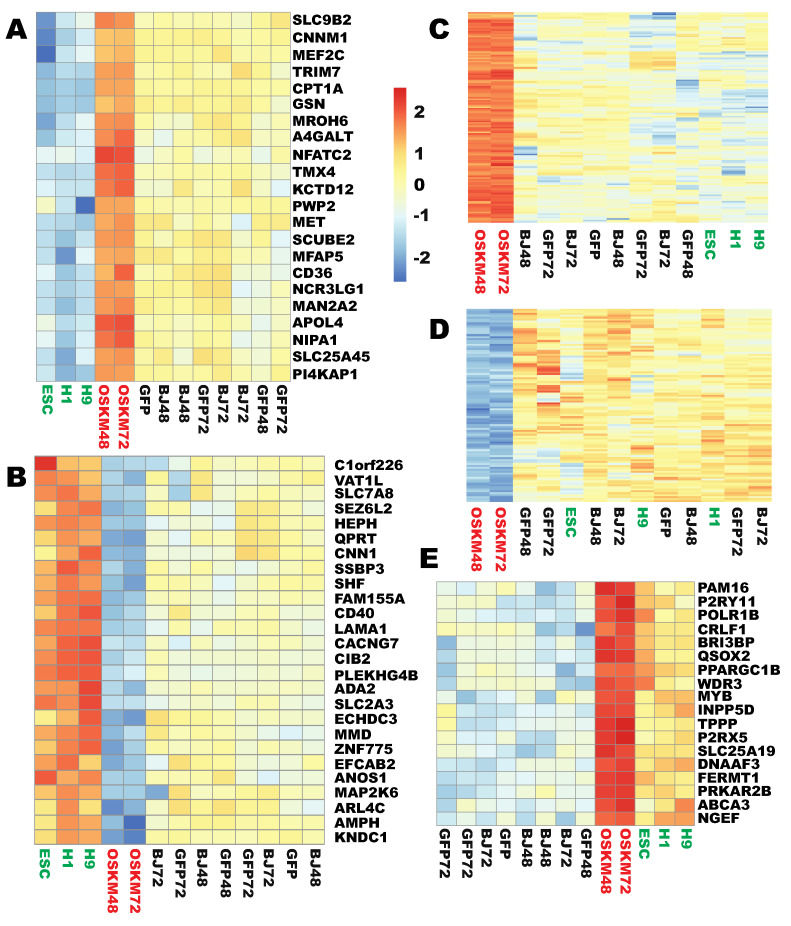
Five types of aberrant reprogramming by OSKM at the early stages. (**A**) A heat map showing that 22 genes were wrongly upreprogrammed. (**B**) A heat map showing that 26 genes were wrongly downreprogrammed. (**C**) A heat map showing that 134 genes underwent unwanted upreprogramming. (**D**) A heat map showing that 99 genes underwent unwanted downreprogramming. (**E**) A heat map showing that 18 genes underwent over-upreprogramming. Figure sample labels are the same as in Figure 1.

**Figure 9 ijms-21-03229-f009:**
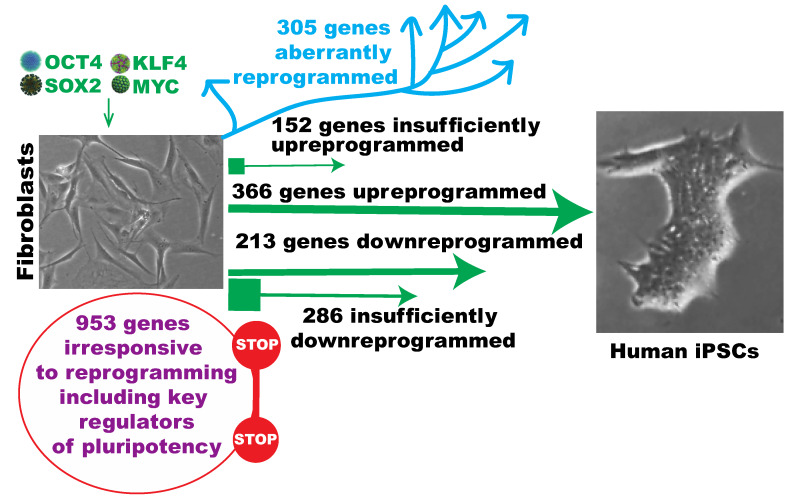
A model for molecular underpinnings of, and resistance to, as well as derailing of Yamanaka reprogramming of human fibroblasts. Green arrows towards the iPSC images indicate positive responses, and the squares at the end of the arrows indicate the reprogramming traction. Cyan arrows away from the iPSC images indicate molecular derailing of reprogramming. Stop signs denote refractory genes of reprogramming as members of the reprogramome.

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
