# Peer review of "A PIANO (Proper, Insufficient, Aberrant, and NO Reprogramming) Response to the Yamanaka Factors in the Initial Stages of Human iPSC Reprogramming"

_ijms, 2020, doi:10.3390/ijms21093229_

Round 1

Reviewer 1 Report

The manuscript titled"  A PIANO.....the initial stages of human iPSC reprogramming" authored by Kejin Hu is a comprehensive study that sheds new insights into the advantages and limitations of the YAMANAKA factor derived reprogramming, especially at the early stages. The author utilized RNA-Seq to unravel the proper, insufficient, and aberrant transcriptional changes that mediated or retracted reprogramming in the early stages. The authors, in line with previous reports, identified a total of 579 'proper reprogramming' and 905 refractory genes to the initial reprogramming. Importantly, the author identified 305 aberrantly expressed genes that derailed and acted as barriers to early reprogramming. In this study, the author has provided compelling data that can lead to improvements in iPSC technology. However, there are a few concerns:

1) Will the expression of the insufficiently reprogrammed genes change over time? Maybe these are not needed at the early stages.

2) Was there any overlap between the insufficiently up- or down-reprogrammed and the aberrantly reprogrammed genes (305)?

Author Response

“Will the expression of the insufficiently reprogrammed genes change over time? Maybe these are not needed at the early stages”

This is a good question, but is not the focus of this study. As I said in the manuscript, to make all the transcriptional responses to OSKM induction reliable, this study focused on the responses that are the same at both time points, 48 and 72 hours post factor transduction. Also, I intentionally choose two close time points (24 hours apart) at the very early stages rather than repeats at the same time points. Such a more orthogonal design of experiments result in more meaningful and reliable data. At these two time points the cells do not show apparent morphology changes based on our experience. However, I do notice that for some genes, the expression levels at the 72 hours is higher. I believe those genes will be further upregulated post 72 hours. However, this is not the focus of the current research. The current research has make several new points already, and provided new insights no other labs have ever done.

Was there any overlap between the insufficiently up- or down-reprogrammed and the aberrantly reprogrammed genes (305)?

No. It is impossible there is any overlap due to the different sorting criteria for those genes. To confirm this, I run quick comparisons for between the 305 genes against the two lists of genes your mentioned, and the software gave zero overlaps for both comparisons.

In addition, I have refined the English. All of the revision in language are minor without extensive re-writing or re-structuring of the sentences. Not significant addition or deletion of text. All changes can be tracked if you will turn on your tracking function to “All Markup”. The followings are some examples of English refinements.

 In line 585, I have changed “difference” to “differences”.

From lines 620 to 622, the part of “while it remains in private status the reviewers can access the data by using this token: yxybussmftchxub. (The data will be available to public after publication of this manuscript)” has been deleted because this was originally provided for the reviewers to use. This is the only significant deletion of text during the revision.

In line 619, one redundant “of” was deleted.

Reviewer 2 Report

Yamanaka reprogramming is revolutionary but inefficient. To understand the molecular mechanism behind that, Dr. Hu expressed the Yamanaka factors (OSKM) in human fibroblasts using lentiviral vectors and systematically analyzed the transcripts obtained 48 and 72 hours after the OSKM expression. The author categorized the transcript into many groups based on their response to OSKM, and performed functional analysis of the genes in each group. The manuscript is very interesting and well written. I recommend publishing this paper in Int. J. Mol. Sci., after correction of typing errors.

Minor points:

  1. References 13 and 15 are the same.
  2. It is better to add the figure legends to Supplementary Figures.
  3. I see both “down-reprogramome” and “downreprogramome (no hyphen)”. It is better to standardize.
  4. Line 458. “contributes” must be “contribute”.
  5. Line 464. “provide” must be “provides”.

Author Response

References 13 and 15 are the same.

This was a mistake that resulted from switching computer during the process of manuscript preparation because of COVID-19 (office computer to home computer because UAB asked employees to work at home). All corresponding changes have been made in the manuscript. All changes can be tracked because I used “track changes” as required. For example, all numbers for references after the reference 15 have been changed accordingly, and the original reference 15 was deleted.

Line 458, “contributes” must be “contribute”.

Yes, this was a typo. Changed.

Line 464, “provide” must be “provides”.

Yes, it was a typo. Changed.

Both “down-reprogramome” and “downreprogramome” are used.

I have revised throughout the manuscript to make the spelling consistent. Please note the no comments were included in the revision, but track of changes can be seen.

It is better to add the figure legends to the supplementary figures.

The legends for both the supplementary figures and tables were provided in the original submission. The editorial staffs may missed those. The legends for the supplementary figures and tables are now attached to the end of the manuscript. It will be appreciated if the editorial staffs could help add these information to each supplementary figures and tables.